# Genetic Characterization of Two Novel Insect-Infecting Negative-Sense RNA Viruses Identified in a Leaf Beetle, *Aulacophora indica*

**DOI:** 10.3390/insects15080615

**Published:** 2024-08-15

**Authors:** Meng-Nan Chen, Zhuang-Xin Ye, Ke-Hui Feng, Jing-Na Yuan, Jian-Ping Chen, Chuan-Xi Zhang, Jun-Min Li, Qian-Zhuo Mao

**Affiliations:** 1State Key Laboratory for Managing Biotic and Chemical Threats to the Quality and Safety of Agro-Products, Key Laboratory of Biotechnology in Plant Protection of Ministry of Agriculture and Zhejiang Province, Institute of Plant Virology, Ningbo University, Ningbo 315211, China; 15066716480@163.com (M.-N.C.); yzx244522794@163.com (Z.-X.Y.); 15290827100@163.com (K.-H.F.); yjn2430304122@163.com (J.-N.Y.); jianpingchen@nbu.edu.cn (J.-P.C.); chxzhang@zju.edu.cn (C.-X.Z.); lijunmin@nbu.edu.cn (J.-M.L.); 2College of Forestry, Nanjing Forestry University, Nanjing 210037, China

**Keywords:** insect-specific virus, negative-sense RNA virus, *Aulacophora indica*, Chrysomelidae, *Nyamiviridae*, *Chuviridae*

## Abstract

**Simple Summary:**

Insect-specific negative-sense RNA viruses are widespread, exhibiting a diverse range of characteristics. Leaf beetles within the Chrysomelidae family, known for their chewing mouthparts, are significant agricultural pests. Despite this, reports of negative-sense RNA viruses in leaf beetles are scarce. In this study, we have identified two novel negative-sense RNA viruses in a globally distributed leaf beetle species, *Aulacophora indica* (Coleoptera, Chrysomelidae). The complete genomes of two insect-infecting viruses (ISVs) were successfully obtained through metatranscriptome analysis and rapid amplification of cDNA ends (RACE), named *Aulacophora indica* nyami-like virus 1 (AINlV1) and *Aulacophora indica* chu-like virus 1 (AIClV1). Phylogenetically, AINlV1 falls into an unclassified clade within the *Nyamiviridae* virus family, while AIClV1 is classified as a member of the *Scarabeuvirus* genus within the *Chuviridae* family. Both viruses exhibit typical genome structures associated with their respective families. Additionally, they activate small RNA interference responses, with virus-derived small RNAs (vsiRNAs) from AINlV1 and AIClV1 showing a consistent pattern peaking at 21 nucleotides, biased towards A/U at the 5′-terminal. Our study provides valuable insights into insect-specific viruses (ISVs) affecting leaf beetles.

**Abstract:**

Herbivorous insects harbor a variety of insect-specific viruses (ISVs) some of which are considered to be valuable biological agents for potential applications in biological defense and control strategies. Leaf beetles with chewing mouthparts are particularly known for their capacity to disrupt plant tissue while feeding, often creating openings that can act as entry points for plant pathogens. In this study, we have identified two new negative-sense RNA viruses infecting the leaf beetle *Aulacophora indica*, an important member of the Chrysomelidae family. These recently discovered viruses belong to the viral families *Nyamiviridae* and *Chuviridae* and have been preliminarily named *Aulacophora indica* nyami-like virus 1 (AINlV1) and *Aulacophora indica* chu-like virus 1 (AIClV1), respectively. The complete genomic sequences of these viruses were obtained using rapid amplification of cDNA ends (RACE) techniques. Detailed analysis of their genomic structures has confirmed their similarity to other members within their respective families. Furthermore, analysis of virus-derived small interfering RNA (vsiRNA) demonstrated a high abundance and typical vsiRNA pattern of AINlV1 and AIClV1, offering substantial evidence to support their classification as ISVs. This research enhances our understanding of viral diversity within insects.

## 1. Introduction

Herbivorous insects host numerous insect-specific viruses (ISVs), which have gained significant attention due to advancements in metagenomics and metatranscriptomics techniques. At the family level, prevalent viral families in agricultural insects include *Iflaviridae*, *Dicistroviridae*, *Partitiviridae*, and *Rhabdoviridae* [1,2]. Insects interact with diverse microbial communities in their environment, establishing a spectrum of relationships ranging from symbiotic to pathogenic. As a result, ISVs are regarded as biologically valuable agents that hold promise for the biological control of insect pests [3,4,5].

The Coleoptera order, encompassing over 360,000 species, constitutes 40% of the world’s known insect species, making it the largest insect order globally [6]. Beetles with chewing mouthparts are known for their capability to disrupt plant tissue while feeding, creating openings that serve as potential entry points for plant pathogens [6,7,8]. Notably, more than 70 beetle species from families such as Chrysomelidae, Coccinellidae, Curculionidae, and Meloidae are recognized as vectors of viruses that infect economically important vegetables and grain crops [6,9,10]. Despite the extensive diversity among beetle species, the discovery of insect-specific RNA viruses associated with them remains limited. The most frequently identified RNA viruses belong to *Iflaviridae*, which is a positive-sense RNA virus family [1]. Negative-sense viruses are integral to the insect RNA virome, with a few sequences been submitted to the NCBI Virus Database; however, limited research has focused on virus characterization.

*Aulacophora indica*, a member of the Galerucinae subfamily within the Chrysomelidae family, is a globally distributed species, predominantly inhabiting tropical regions [11,12]. Adult leaf beetles primarily feed on foliage, targeting the parenchyma of the lower leaf surface and leading to the formation of irregular holes. While leaf beetles are known to transmit several significant plant viruses, the identified ISVs have been scarce [7,9,13]. *A. indica* has a close association with *Aulacophora lewisii*, from which a novel Aulacophora lewisii iflavirus 1 (ALIV1) was identified [14], while ISVs infecting *A. indica* remain unidentified.

In this study, two novel leaf beetle-infecting negative-sense RNA viruses, named *Aulacophora indica* nyami-like virus 1 (AINlV1) and *Aulacophora indica* chu-like virus 1 (AIClV1), were discovered in *A. indica* using transcriptomic analysis. The presence of ISVs was confirmed through RT-PCR, and their complete genome sequences were validated using rapid amplification of cDNA ends (RACE). Additionally, analysis of virus-derived small interfering RNA (vsiRNA) clearly indicated an active antiviral RNA interference (RNAi) pathway in *A. indica* in response to these two viruses.

## 2. Materials and Methods

### 2.1. Sample Preparation and RNA Extraction

A single adult leaf beetle was collected in August 2022 from the leaves of a melon in Ningbo, Zhejiang Province, China. Total RNA was subsequently extracted using TRIzol reagent (Invitrogen, Carlsbad, CA, USA) following the manufacturer’s instructions. The quality of the extracted RNA was confirmed using a NanoDrop spectrophotometer (ThermoFisher Scientific, Madison, WI, USA). Subsequently, the extracted RNA samples were used for transcriptomic sequencing, small RNA (sRNA) sequencing, and full genome sequence determination.

### 2.2. Transcriptome and sRNA Sequencing

The total RNA samples were sent to Novogene (Tianjin, China) for transcriptome and small RNA sequencing. As previously described, libraries were prepared for transcriptome and sRNA analysis [15]. In brief, for transcriptome sequencing, paired-end (150 bp) libraries were constructed and sequenced on the Illumina HiSeq 2500 platform (Illumina, San Diego, CA, USA). The raw reads were quality-trimmed for adapter sequences using Trinity software (Version 2.8.5) with default parameters, followed by de novo assembly of clean reads [16]. For sRNA sequencing, we sequenced on the Illumina HiSeq 2500 platform and prepared an sRNA library using the Illumina TruSeq Small RNA Sample Preparation Kit (Illumina, San Diego, CA, USA). The outputted raw data were trimmed by removing adapters and low-quality sequences using the Cutadapt tool [17].

### 2.3. Host Insect Identification

To identify the exact species of leaf beetle, the assembled transcript contigs were compared with all the available cytochrome oxidase subunit 1 (COI) barcode records from the Barcode of Life Data (BOLD) Systems (http://www.boldsystems.org/, accessed on 24 February 2024) and the National Center for Biotechnology Information (NCBI) using Blastn. The identified COI sequence was cloned and further confirmed using Sanger sequencing. The primers used for COI amplification are listed in Appendix A.

### 2.4. Virus Discovery and Confirmation 

The potential viruses were identified in the samples using the previously described method. Briefly, the assembled RNA-seq clusters were compared to the NCBI virus RefSeq database using BlastX, with an E-value cutoff set at 1 × 10^−20^ [18]. To eliminate false-positive matches, the viral homology contigs exceeding 1000 bp in lengths were chosen for Blastn and Blastx analysis against the NCBI complete nucleotide (NT) and non-redundant (NR) protein databases. Finally, the identified potential viral contigs were further validated using RT-PCR and Sanger sequencing using the primers listed in Appendix A.

### 2.5. Determination of Virus Full Length Genome and Transcript Abundance

To obtain the full length of the two identified viruses in the insect sample, the extreme 5′ and 3′ terminal sequences were determined using rapid amplification of cDNA ends (RACE) with SMARTer^®^ RACE 5′/3′ kit (Takara, Beijing, China). After total RNA extraction, first-strand cDNA synthesis was performed to obtain 5′-RACE-ready and 3′-RACE-ready cDNA according to the manufacturer’s instructions. Touchdown PCR was performed to amplify RACE products using 5′ or 3′ GSPs (gene-specific primers) and UPM (Universal Primer A Mix). The PCR reaction (50 μL) included a 1.5 μL cDNA Adaptor, 1.5 μL 10 × UTM primer mix, 1.5 μL GSPs, 25 μL 2 × Phanata Buffer, 1 μL dNTP, and 1 μL Phanata Max Super-Fidelity DNA polymerase (Vazyme, Nanjing, China), with the addition of 20 μL ddH_2_O. The PCR thermal cycling protocol was as follows: denaturation at 95 °C for 3 min, followed by 10 cycles of 95 °C for 30 s, 65 °C for 30 s, and 72 °C for 30 s, subsequently followed by 30 cycles of 95 °C for 30 s, 60 °C for 30 s, and 72 °C for 30 s. The primers used for RACE are listed in Appendix A. Later, the PCR products were cloned into the pClone007 vector (Tsingke, Beijing, China), followed by Sanger sequencing.

To investigate the transcript coverage and the viral abundance, the transcriptome reads were trimmed for adapters and for quality through Bowtie2 (v2.3.5.1) and Samtools (v1.7) software [19] and were aligned against the complete viral genome. The matched reads coverage was visualized using an Integrated Genomics Viewer [20].

### 2.6. Viral Genome Annotation and Phylogenetic Analysis

The open reading frames (ORFs) of viruses were predicted using TBtools-II (Toolbox for Biologists) v2.069 [21]. Additionally, the conserved protein domains were predicted using the InterProScan (https://www.ebi.ac.uk/interpro, accessed on 14 March 2024) and HHpred (https://toolkit.tuebingen.mpg.de/tools/hhpred, accessed on 14 March 2024).

For phylogenetic analysis, the predicted RNA-dependent RNA polymerase (RdRP) domain of the novel viruses combined with RdRP protein sequences from reference viruses were used. For reference viruses, we selected the representatives from viral families *Nyamiviridae*, *Lispiviridae*, *Aliusviridae*, and *Chuviridae*, and viruses showing similar identity with the two novel viruses from NCBI. The phylogenetic tree was built using PhyloSuite v1.2.3 [22]. The RdRP amino acid sequences were first aligned using MAFFT, and then the aligned sequences were trimmed using trimAl [23,24]. IQ-TREE v2.2.0 was used to infer maximum likelihood phylogenies under the LG + R5 + F model using 1000 ultrafas [25] bootstraps. We next assessed tree-building models in ModelFinder. Finally, iTOL was used to visualize the tree [25]. Detailed virus RdRP sequences are listed in Appendix A. MegAlign (version 7.1.0) and BioEdit (version 7.1.11) were utilized to compare the RdRP protein sequences of the two newly discovered viruses in our study with those of their corresponding related viruses [26]. 

### 2.7. Small RNA Analysis

To identify the virus-derived siRNAs (vsiRNAs), the 18–30 nt long quality-trimmed sRNA reads were extracted using FASTX-Toolkit (http://hannonlab.cshl.edu/fastx_toolkit/, accessed on 18 March 2024). The processed reads were mapped back to the assembled complete genome of the novel viruses using the zero-mismatch Bowtie software [19,27]. The downstream analysis of the output vsiRNAs was performed using custom Perl scripts and Linux bash scripts, including the number and size distribution of the vsiRNA, the distribution of the vsiRNA along the corresponding viral genome, and the 5′-terminal nucleotide preference of the vsiRNA.

## 3. Results

### 3.1. Transcriptome Assembly and Virus Identification in A. indica

A total of 125,744 contigs were generated from the de novo assembly of the clean RNA-seq reads (23,543,893), with the N50 value of 1582 bp. Subsequent BLAST analysis of the COI sequences confirmed the identity of the leaf beetle species as *Aulacophora indica*, exhibiting a significant 99.21% similarity to the previously reported COI sequences of *Aulacophora indica* (Accession Number: NC_047467.1).

To identify potential viruses, the assembled contigs underwent BLAST searches against the NCBI nucleotide (nt) and viral reference databases. This search led to the discovery of two previously unidentified RNA viruses from the transcriptome data of *A. indica*. The near-complete genomes of these viruses were obtained. Based on the taxonomic relations with the most closely associated viruses, both contigs were classified as negative-sense RNA viruses. One contig of 9802 nt was most similar with the Ixodes ricinus orinovirus-like virus 1 (Acession Number: ON684369.1) with protein sequence identities of 47.12% and was categorized as a putative member of the *Nyamiviridae* virus family. The other contig of 9009 nt displayed the closest similarity to the Hangzhou altica cyanea chuvirus 1 (Accession Number: MZ209708.1), with protein identities of 45.46%, and was identified as a prospective member of the *Chuviridae* virus family. Consequently, they were tentatively named “*Aulacophora indica* Nyami-like virus 1” (AINlV1) and “*Aulacophora indica* chu-like virus 1” (AIClV1), respectively.

### 3.2. Genome Structure of Two Novel Negative-Sense RNA Viruses

The presence of two novel negative-sense RNA viruses, AINlV1 and AIClV1, was confirmed through RT-PCR followed by Sanger sequencing. The complete genome sequences were obtained by RACE to identify the 5′ and 3′ termini using the SMARTer^®^ RACE 5′/3′ Kit. The full genome sequence of AINlV1 was 9873 nt in length (excluding polyA) (GenBank accession number: PP888187) and comprised six predicted open reading frames (ORFs), a 90-nt 3′UTR, and a 56-nt 3′UTR (Figure 1A). InterProScan prediction (https://www.ebi.ac.uk/interpro, accessed on 14 March 2024) of conserved domains indicated that ORF1 (91–1077 nt) contained a nucleoprotein domain, ORF5 (2674–4101 nt) contained a glycoprotein domain, and ORF6 (4185–4955 nt) contained an RNA-directed RNA polymerase (RdRP) catalytic domain and an mRNA-capping domain (Figure 1A). It also indicated that ORF I, ORF V, and ORF VI encoded the nucleocapsid (N) protein, glycoprotein (G), and large (L) protein, respectively, which are basic and critical for viruses from *Nyamiviridae*. The functions of ORF II, ORF III, and ORF IV remain undetermined. Assessment of AINlV1 abundance and coverage involved aligning RNA-seq sequences with the reconstructed complete genome to reveal a mean coverage of 147. The distribution of transcripts throughout the viral genome indicated the efficient replication of AINlV1 in the host insect (Figure 1A).

Regarding the novel chu-like virus AIClV1, the complete genome was 11,590 nt (GenBank accession number: PP888188), housing four predicted ORFs. “Around-the-genome RT-PCR” from the 3′ end to the 5′ end illustrated the circular form of this viral sequence (Figure 1B,C). Through the identification of conserved domains using InterProScan and HHpred, it was deduced that ORF I (70–6708 nt) encoded an RdRP catalytic domain and an mRNA-capping domain (L protein), ORF II (7242–9272 nt) encoded a glycoprotein domain (G protein), ORF III (9631–10,971 nt) encoded a nucleoprotein domain (N protein), and ORF IV (11,079–11,528 nt) encoded a protein with an unknown function. This “L-G-N-(x)” structure was consistent with previously reported circular chuviruses [28,29]. A realignment of RNA-seq reads to the reconstructed full genome sequence of AICV1 revealed a relatively abundant distribution of transcripts across the entire viral genome, with a mean coverage of 50 (Figure 1B).

### 3.3. Phylogenetic Analysis of Novel Viruses

To further elucidate the taxonomical classification of the two novel negative-sense RNA viruses, a phylogenetic analysis was performed based on the RdRP protein sequences. This analysis utilized reference virus sequences from *Nyamiviridae*, *Lispiviridae*, *Aliusviridae*, and *Chuviridae*. The amino acid sequences of RdRPs were aligned, and a maximum likelihood (ML) tree was constructed. The results of the phylogenetic analysis indicated that AINlV1 clustered with Ixodes ricinus orinovirus-like virus 1 (USL85442.1, identified in tick, *Ixodes ricinus*) in an unclassified clade, with a relatively high bootstrap value (Figure 2A,B). The RdRP sequence of AINlV1 exhibited 26% to 40% homology with viruses from the *Fomivirus* and *Orinovirus* genera, suggesting that AINlV1 represents a novel member of the *Nyamiviridae* family. In terms of AIClV1, the phylogenetic analysis revealed a close clustering with Lishi spider virus 1 (YP_010839348.1, identified in spider) and Hangzhou altica cyanea chuvirus 1 (UHR49734.1, identified in bug, *Zicrona caerulea*) within the *Scarabeuvirus* genus of *Chuviridae* (Figure 2A,C). The similarity of AIClV1 to other scarabeuviruses ranged from 31% to 42% for RdRP sequences. The close phylogenetic relationship with ISVs suggested that these two new viruses were ISVs.

### 3.4. Activation of the Antiviral RNA Interference Pathway in A. indica in Response to Novel Viruses

The RNA interference (RNAi) pathway mediated by small interfering RNA (siRNA), serves as a pivotal innate antiviral defense mechanism in insects. This pathway is activated upon viral invasion and proliferation [30,31]. To gain deeper insights into the siRNA-based antiviral mechanisms of leaf beetles in countering negative-sense RNA viruses, we conducted small RNA (sRNA) sequencing on the leaf beetle *A. indica*. We characterized the virus-derived siRNAs (vsiRNAs) of AINlV1 and AIClV1. These vsiRNAs demonstrated a distribution pattern peaking at 21 nt and originated equally from both the sense and antisense strands of the viral genome (Figure 3A,D). Notably, the 21 nt vsiRNAs exhibited a pronounced A/U bias in their 5′-terminal nucleotides (Figure 3B,E), a common characteristic observed in vsiRNAs among insect species [32,33]. These vsiRNAs were uniformly distributed across the two novel viruses, with specific regions showing a preference for Dicer-mediated targeting (Figure 3C,F). The distinctive vsiRNA profile observed suggests an active involvement of the host antiviral RNAi pathway in responding to viral infections.

## 4. Discussion and Conclusions

Several beetle members of the family Chrysomelidae have been discovered to transmit plant viruses; however, only a limited number of insect-specific viruses (ISVs) have been documented [6,7,10,12,34]. This study focused on identifying and characterizing two novel ISVs obtained from *A. indica*, a leaf beetle species. By utilizing the rapid amplification of cDNA ends (RACE) technique, we successfully isolated and sequenced the complete genomes of two negative-sense RNA viruses. A thorough phylogenetic analysis determined that these two viruses belong to the families *Nyamiviridae* and *Chuviridae*, respectively. Further analysis of their genome structures confirmed their resemblance to other members within their respective families. Notably, the abundance and typical pattern of vsiRNAs associated with these viruses offer compelling evidence supporting their classification as ISVs.

Numerous insect-borne negative-sense RNA viruses have garnered attention due to their pathogenic impact on humans, animals, and plants [3,35,36]. Recent research has demonstrated that insect-specific negative-sense RNA viruses are not only widespread but also exhibit diverse characteristics [3,35]. The *Nyamiviridae* family, which falls under the *Mononegavirales* order, encompasses several genera, including *Nyavirus*, *Socyvirus*, *Berhavirus*, *Crustavirus*, *Orinovirus*, *Tapwovirus*, and *Formivirus*. Nyamiviruses possess a wide host range, infecting both invertebrates and certain vertebrates [37,38,39,40]. For instance, Nyamanini virus (NYMV) and Midway virus (MIDWV) are tick-borne viruses that infect land birds and seabirds, respectively [41]. Soybean cyst nematode virus 1 (SbCNV-1) was detected in soybean cyst nematodes [42]. Moreover, specific members of the *Formivirus* genus have demonstrated associations with ants and wasps [38,43]. In our investigation, we identified a novel nyamivirus, named *Aulacophora indica* nyami-like virus (AINlV1), which grouped together with Ixodes ricinus orinovirus-like virus 1 (GenBank Accession Number: ON684369.1) within an unclassified clade (Figure 2A,B), indicating the potential existence of a new genus. Initially identified in various arthropods through metagenomic assessments as “chuviruses”, *Jingchuvirales* were initially classified as part of the *Mononegavirales* virus order based on phylogenetic analyses [44,45]. However, owing to their distinct genomic structures and phylogenetic positions, a novel order known as *Jingchuvirales* was officially established in 2022 [29,44]. *Jingchuvirales* exhibit diverse genome structures, including non-segmented, segmented, and possibly circular arrangements, differing from the conventional *Mononegavirales* order. Furthermore, *Jingchuvirales* are now recognized to have a broad distribution across multiple animal phyla [46,47,48,49]. In our analysis, the chu-like virus AIClV1 identified in *A. indica* demonstrated a circular genome sequence form and a typical “L-G-N-(x)” gene order, thus being categorized in the genus *Scarabeuvirus*. Given the extensive host range of nyamiviruses and chuviruses, further research is required to explore potential hosts for these two newly discovered viruses.

The antiviral immune response mediated by small RNAs (sRNAs) is a crucial innate pathway in insects. The generation and distinct attributes of viral small interfering RNAs (vsiRNAs) play a vital role in determining a virus’s capacity to infect insects [31,50]. The characteristics of vsiRNAs, including size, polarity, and base preferences, elucidate the interplay between host antiviral mechanisms and viral RNA, offering valuable insights into virus biology and immune evasion strategies [33,51]. In hemipterans like planthoppers, vsiRNAs are primarily composed of 21 and 22 nucleotides (nt) [17], whereas dipterans, such as *Drosophila melanogaster*, exhibit a predominant vsiRNA pattern at 21 nt [33]. Notably, there is currently a lack of reports on vsiRNA characteristics within the leaf beetles of the Chrysomelidae family. This study delineated the vsiRNA profiles of two negative-sense RNA viruses identified in *A. indica*. These results indicate that the vsiRNA profiles of these viruses primarily display a peak at 21 nt and are evenly distributed across both the sense and antisense strands (Figure 3), which is in line with the vsiRNA pattern observed in other insect species.

The study unveiled the complete genome sequences of two novel negative-strand RNA viruses within *A. indica*. This revelation marks the initial recognition of negative-strand RNA viruses in *A. indica*, augmenting our understanding of viral diversity among insects. Further research is crucial to understand the distribution and transmission of viruses among leaf beetles in their natural habitats, as well as to investigate the pathogenicity and impact of these viruses on insect physiology. These investigations are fundamental for evaluating the potential effectiveness of utilizing viruses for biocontrol purposes.

## Figures and Tables

**Figure 1 insects-15-00615-f001:**
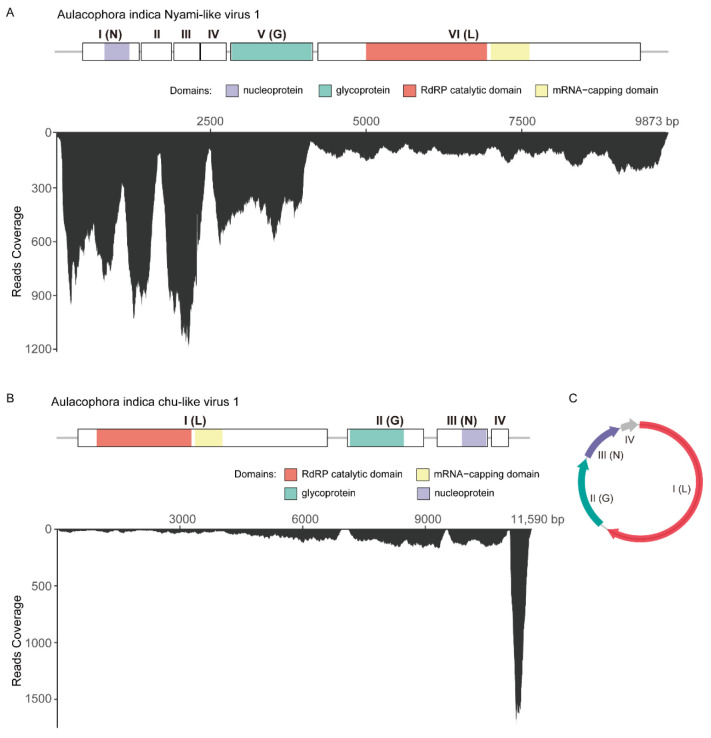
Genome structure and RNA-seq read coverage of two negative-sense viruses, AINlV1 and AIClV1. (**A**) The genome organization of AINlV1 shows that this virus contains six open reading frames (ORFs), with ORF I, ORF V, and ORF VI encoding N, G, and L protein, respectively. The mean coverage of RNA-seq reads was 147. (**B**,**C**) The viral sequence of AIClV1 is in a circular form (**C**) and contains four ORFs, with ORF I, ORF II, and ORF III, encoding L, G, and N protein (**B**), which is consistent with the typical structure of “L-G-N-(x)” of chuviruses. The mean coverage of RNA-seq reads to this virus was 50.

**Figure 2 insects-15-00615-f002:**
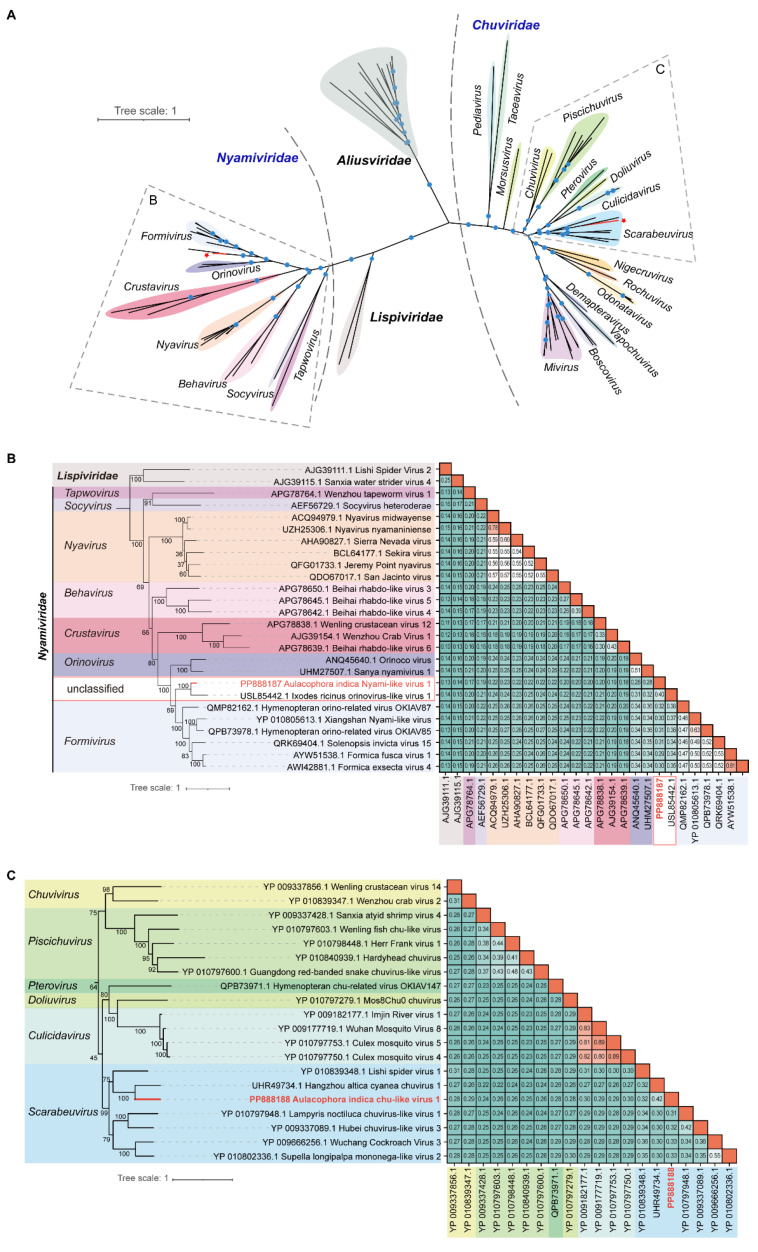
Phylogenetic analysis and RdRP protein identities of the two negative-sense RNA viruses with others. (**A**) Maximum likelihood phylogenetic tree based on the RNA-directed RNA polymerase domain was constructed with a bootstrap of 1000. (**B**) Phylogenetic analysis of AINlV1 in family *Nyamiviridae* (**left**), and identities (%) of amino acid sequences of AINlV1 with other viruses from family *Nyamiviridae* (**right**). (**C**) Phylogenetic analysis (**left**) and amino acid sequence identities (%) (**right**) of AIClV1 in family *Chuviridae*.

**Figure 3 insects-15-00615-f003:**
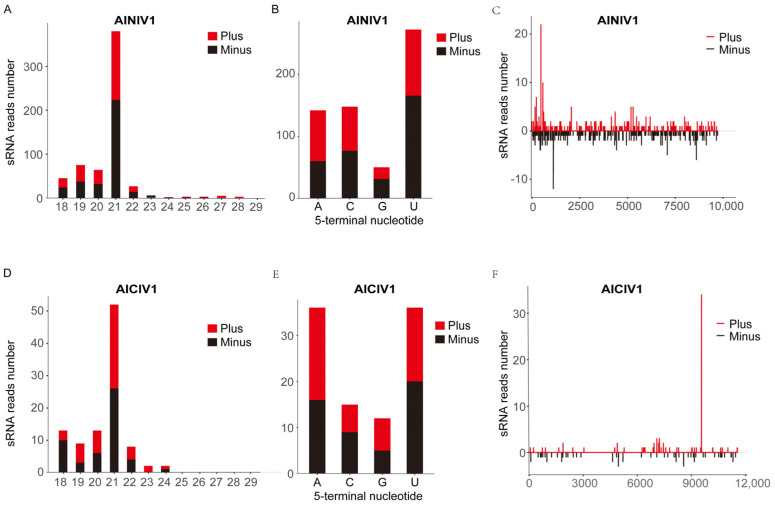
Profiles of virus-derived small interfering RNAs (vsiRNAs) of *Aulacophora indica* Nyami-like virus 1 (AINlV1) and *Aulacophora indica* chu-like virus 1 (AIClV1). (**A**) Size distribution of AINlV1-derived sRNAs. (**B**) The 5′-terminal nucleotide preference of the 21-nt long vsiRNAs of AINlV1. (**C**) Distribution of vsiRNAs corresponding to AINlV1 genome. (**D**) Size distribution of AIClV1-derived sRNAs. (**E**) The 5′-terminal nucleotide preference of 21-nt long vsiRNAs of AINlV1. (**F**) Distribution of vsiRNAs corresponding to AIClV1 genome.

## Data Availability

The viral nucleotide sequences were uploaded to NCBI with GenBank accession numbers ON684369.1 and MZ209708.1. The authors declare that all data supporting the findings of this study are available in the manuscript and its Appendix A are available from the corresponding authors upon request.

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
