# Peer review of "Genetic Characterization of Two Novel Insect-Infecting Negative-Sense RNA Viruses Identified in a Leaf Beetle, Aulacophora indica"

_insects, 2024, doi:10.3390/insects15080615_

Round 1

Reviewer 1 Report

Comments and Suggestions for Authors

Dear authors

The submitted manuscript describes the characterisation of two negative-sense RNA viruses. These were identified in a single leaf beetle sample, from the species Aulacophota indica. This was acheived via RNA sequencing of both total RNA and small RNA enabling host determination via COI analysis, virus identification, and identification of virus-derived siRNAs.

The methodology utilised and findings reported in the manuscript are suitable, with the results indicating the identification of two novel viruses. The mansuscript provides a simple process for the identification of these novel viruses and provides some insight into the antiviral RNA interference pathway of the host insect based on the sequencing of virus-derived siRNAs.

The corrections provided are only minor:

Title: The authors state in the title that the identified viruses are insect specific, however, this is not evlauted in the manuscript. According to the ICTV report on Nyamiviridae, members in this family are associated with invertebrates as well as land- and seabirds. While the genera these viruses clustered with in the phylogenetic analysis are invertebrate infecting (e.g. Formivirus with Ants and wasps and Orinovirus with moths), the host range was not fully identified. Similarly, members in the Jingchuvirales order are reported to have a host range extending into fish and reptiles. AICLV1 was identified to group most closely with members of the genus Chuviridae, which are are descrived in the ICTV report to have a host range including invertebrates, fish, and snakes. Given this, I am not confident one can indicate that the two novel viruses are indeed insect specific.

Line 19: The authors use the abbreviation ISV without providing the full term.

Line 89: Suggested to write as "... the extracted RNA samples underwent transcriptomic sequencing, small RNA ..."

Line 100: The quality is not trimmed, the sequences are trimmed to remove adapters and low-quality sequences to improve overall quality.

Line 114: Correct spelling of "vial" to "viral"

Line 127: Subscript in ddH2O

Line 135: Missing space between "(V1.7) software"

Line 150: Missing space after ultrafas

Line 151: The authors switch between RdRP and RdRp, I believe the former is correct and this must be adjusted for consistency across the paper.

Line 220 and 223: Past tense, "reads was 147" and "this virus was 50."

Line 226: RdRP already abbreviated early, no need to repeat the abbreviation.

Line 258 and 260: There is no Figure 4 reported, these should be corrected to Figure 3.

Kind regards

Comments on the Quality of English Language

The quality of English is suitable, there are some minor typing errors whic need to be corrected. The manuscript was otherwise clear and easily understood.

Author Response

We appreciated your insightful comments and suggestions on this paper, as these comments led us to an improvement of the work. Our revisions reflect all the suggestions and comments. 

Reviewer 2 Report

Comments and Suggestions for Authors

Chen et al. report the identification of two novel RNA viruses from a chrysomelid beetle pest. The study appears to have been correctly performed and it is clear that the authors have a good understanding of the current taxonomy of these viruses. I have only minor suggestions for improvements:
I have written suggestions and numbered points on a scanned copy of the manuscript.
Numbered points (see scanned file)
1. Chrysomelinae is a SUB-family of the Chrysomelidae. This needs to be corrected throughout the manuscript.
2. For clarity, write out the meaning of 'ISV' in the summary and Abstract and at first use in the body of the text.
3. The word 'genus' should not be italicized.
4. You have no evidence for pathogenicity or biocontrol potential for the novel viruses. Please delete this text.
5. Please provide the primer sequences that you used for COI amplification (I could not find them in Table S1).
6. The authors need to recognize that they found the present viruses in a single individual beetle and did not assess pathogenicity, virulence or sublethal disease effects in any way. The conclusion of the study should point this out. Only after such studies would it be possible to opine on the biocontrol potential of the viruses.
7. Title: The viruses have been genetically characterized, but no biological characterization (phenotypical traits) has been performed at all. I think that the term "Characterization" should be removed or changed to "Genetic characterization".
Finally, the references should be formatted according to journal guidelines (I only checked the first page).

Comments on the Quality of English Language

Well written, minor editing.
